# A Comprehensive Study on the Nutritional Profile and Shelf Life of a Custom-Formulated Protein Bar Versus a Market-Standard Product

**DOI:** 10.3390/foods14122141

**Published:** 2025-06-19

**Authors:** Corina Duda-Seiman, Liliana Mititelu-Tartau, Simona Biriescu, Alexandra-Loredana Almășan, Bianca-Oana Bitu, Adina-Ioana Bucur, Andrei Luca, Bogdan Hoinoiu, Teodora Hoinoiu

**Affiliations:** 1Department of Cellular and Molecular Biology, Faculty of Medicine, Titu Maiorescu University, Văcărești 187 Street, 031593 Bucharest, Romania; corina.duda@e-uvt.ro (C.D.-S.); lc_andrei@yahoo.com (A.L.); 2Department of Pharmacology, Clinical Pharmacology and Algesiology, Faculty of Medicine, “Grigore T. Popa” University of Medicine and Pharmacy, Universitatii 16 Street, 700115 Iasi, Romania; 3Department of Finance, Information Systems and Business Modeling, Faculty of Economics and Business Administration, West University of Timisoara, Johann Heinrich Pestalozzi 16 Street, 300115 Timisoara, Romania; simona.biriescu@e-uvt.ro; 4Department of Anatomy, Animal Physiology and Biophysics, Faculty of Biology, University of Bucharest, Splaiul Independentei 91-95 Street, 050095 Bucharest, Romania; alexandra.almasan1998@gmail.com; 5Department of Biochemistry, Faculty of Chemistry-Biology-Geography, West University of Timisoara, Johann Heinrich Pestalozzi 16 Street, 300115 Timisoara, Romania; bianca.bitu01@e-uvt.ro; 6Department of Functional Sciences, Discipline of Public Health, Center for Translational Research and Systems Medicine, “Victor Babes” University of Medicine and Pharmacy, Eftimie Murgu Square 2, 300041 Timisoara, Romania; bucur.adina@umft.ro; 7Department of Oral Rehabilitation and Dental Emergencies, Faculty of Dentistry, “Victor Babes” University of Medicine and Pharmacy, Eftimie Murgu Square 2, 300041 Timisoara, Romania; hoinoiu@umft.ro; 8Interdisciplinary Research Center for Dental Medical Research, Lasers and Innovative Technologies, “Victor Babes” University of Medicine and Pharmacy, Eftimie Murgu Square 2, 300041 Timisoara, Romania; 9Department of Clinical Practical Skills, “Victor Babes” University of Medicine and Pharmacy Timisoara, 300041 Timisoara, Romania; tstoichitoiu@umft.ro; 10Center for Advanced Research in Cardiovascular Pathology and Hemostaseology, “Victor Babes” University of Medicine and Pharmacy, Eftimie Murgu Square 2, 300041 Timisoara, Romania

**Keywords:** protein, protein bars, macronutrients, soy

## Abstract

Background: With growing interest in healthy lifestyles, protein bars have gained popularity. However, many commercial bars contain excessive calories, sugar, and artificial additives that undermine their health benefits. This study aimed to develop a protein bar using natural ingredients with a balanced macronutrient profile. Method: The protein bar formulation used soy protein extract, a plant-based protein source, known for its complete amino acid profile but limited in methionine, which was complemented by oats to nutritionally balance this deficiency. A database was created to evaluate the cost-effectiveness of commercially available protein bars based on consumer feedback. The experimental bar was tested for nutritional value, shelf life, and physiological impact, using only natural ingredients for texture, flavor, and stability. Results: The experimental protein bar had higher protein and fiber content than a selected commercial bar but a shorter shelf life (7 days vs. 90 days) due to the absence of preservatives. The database helped identify target consumer groups and ensure the product was affordable and nutritionally effective. Conclusion: This study demonstrates that using natural, complementary ingredients can create a protein bar with a more balanced nutrient profile while avoiding harmful additives. The final product supports muscle protein synthesis through its high-quality protein content and promotes glycemic control and satiety via its fiber-rich, low-sugar formulation and metabolic processes, offering a healthier alternative to commercial options, with a focus on consumer health and cost-effectiveness.

## 1. Introduction

In today’s rapidly evolving market, consumer behaviors are constantly shifting, directly influencing lifestyle choices [1]. One significant trend is the increasing focus on health and wellness, driving people to adopt habits such as regular exercise, proper hydration, and healthier eating patterns. In response to this demand, food manufacturers have developed products with higher protein content while reducing fat and calorie levels, catering to consumers’ desire for better nutrition [2].

With the rising popularity of health-conscious eating, protein bars have gained significant recognition. First introduced in the early 1980s, these bars were initially consumed mainly by fitness enthusiasts and professional athletes seeking a convenient source of nutrition to support their performance [1]. Over time, their appeal has expanded, and they have become a staple in many diets. Today, protein bars play a key role in various markets, including sports nutrition, muscle development, weight management, and general health supplementation. Their success stems from their ability to provide a complex array of essential nutrients in a convenient, portable form [3].

The continuous search for innovative food products that deliver essential micro- and macronutrients in an easy-to-consume and efficient manner has further driven the development of such products [4]. Nutrition bars are formulated to contain a balanced mix of proteins, carbohydrates, fats, vitamins, and minerals necessary for daily nutrition. These products are particularly beneficial for individuals engaged in sports and fitness, as they help support muscle growth, enhance recovery, and replenish energy lost during workouts [5,6].

Dietary supplements, including protein bars, are essential for those looking to optimize their physical performance while maintaining proper nutrition. Whether aiming to build muscle, lose weight, or simply meet daily nutritional requirements, these products offer a practical and effective solution to support overall health and wellness [5,6].

Many protein-rich bars derive their protein content from dairy-based sources, such as whey and casein, or from plant-based alternatives, particularly legumes. Among plant proteins, soy stands out due to its high biological value and well-balanced amino acid profile, which closely resembles that of high-quality whey protein [7]. The incorporation of soy protein in protein bars not only provides a cost-effective formulation but also allows manufacturers to emphasize its scientifically recognized cardiovascular benefits on product labels, making it an attractive option for health-conscious consumers [8].

Soy protein is particularly valued for its isoflavone content, bioactive compounds linked to various health benefits, including the reduction of LDL cholesterol levels. These properties contribute to its reputation as a heart-healthy ingredient and have made it a preferred choice in the development of functional foods [9]. Additionally, protein bars are frequently fortified with essential vitamins and minerals, enhancing their nutritional value and making them particularly beneficial for individuals suffering from malnutrition or those with increased dietary requirements.

Despite their nutritional advantages, ongoing debates persist regarding the overall composition of protein bars and their physiological impact. The balance and interaction of macronutrients within these bars play a crucial role in determining their effects on metabolism, muscle synthesis, and overall health. As a result, our research, informed by an extensive review of the scientific literature, aims to develop a protein bar based on soy protein extract that delivers a proportionally balanced intake of essential nutrients.

Current studies on protein bars explore multiple aspects of their formulation and functional properties. Key areas of focus include the influence of protein sources on bar texture, the physical and mechanical interactions between ingredients during manufacturing [10], advancements in formulation techniques to enhance sensory appeal and shelf stability [3,6,11], and the effects of protein bar consumption on specific biochemical parameters [12,13]. One of the most persistent challenges in protein bar development is the tendency of these products to harden over time, leading to undesirable changes in texture that compromise palatability and consumer acceptance [8]. This phenomenon has been well documented in recent studies, including Jiang et al., which identified protein–polysaccharide interactions and moisture migration as key contributors to bar hardening during storage [5,14,15].

While prior studies, such as Małecki et al., have examined the impact of protein source and syrup type on texture and stability in soy-based bars [10], our study advances this work by formulating a preservative-free, cold-processed soy protein bar using only natural ingredients. Additionally, we offer a direct comparison to a commercial counterpart in terms of nutritional value, shelf life, and cost-effectiveness, emphasizing practical applicability and clean-label formulation. Addressing this issue remains a critical objective in the ongoing refinement of protein bar formulations, ensuring both nutritional efficacy and an enjoyable eating experience.

In this research, we aimed to develop a protein bar with a well-balanced macronutrient profile, utilizing natural ingredients to enhance nutritional value while minimizing potential harmful effects associated with artificial additives, excessive sugar, and unhealthy fats. Our objective was to create a cost-effective and accessible product that meets consumer dietary needs, supports overall health, and aligns with modern trends in functional foods. Additionally, we sought to optimize ingredient selection to improve the bar’s texture, taste, and stability without compromising its nutritional integrity.

## 2. Materials and Methods

### 2.1. Protein Source Selection and Soy Protein Extraction

Soy protein is widely recognized as one of the few plant-based proteins that provides a complete profile of essential amino acids, including lysine, leucine, and valine, typically found in high concentrations in animal proteins [16]. Due to its balanced amino acid composition, soy protein is considered a complete protein source according to the Food and Agriculture Organization (FAO) standards [17]. Furthermore, it is an ideal alternative for various dietary groups, notably vegetarians, vegans, and individuals with lactose intolerance, offering numerous health benefits such as improved lipid profiles, enhanced antioxidant activity, and potential cardioprotective effects.

Despite its many advantages, soy protein does have some limitations. It is often considered to be of lower quality than animal-based proteins due to a deficiency in methionine, one of the essential amino acids. This deficiency can impact the efficiency of protein synthesis, as methionine is crucial for the proper utilization of other amino acids. However, this issue can be addressed through the concept of mutual integration of plant-based proteins. To address this amino acid deficiency and achieve a more balanced protein profile, oats may be incorporated as a complementary ingredient, as they provide methionine and help complete the essential amino acid spectrum when combined with soy protein. Mutual integration involves combining complementary plant proteins to ensure that the deficient amino acids of one protein source are supplied by another. In the case of soy protein, the complementary source is often cereal proteins, which are rich in methionine [18]. By combining soy protein with cereal-based proteins, the overall amino acid profile is balanced, enhancing the biological value of the protein and making it more comparable to animal-based protein sources.

The concept of mutual supplementation between soy and oat proteins is supported by protein quality indices such as PDCAAS (Protein Digestibility-Corrected Amino Acid Score) and DIAAS (Digestible Indispensable Amino Acid Score). Soy protein isolate has a high PDCAAS (typically 0.91–1.00), but methionine is its limiting amino acid. Oat protein, although lower in total protein content, is relatively rich in methionine and cysteine, which enhances the overall amino acid balance when combined with soy [17]. According to FAO/WHO guidelines, combining legume and cereal proteins in complementary ratios can improve the digestibility and biological value of plant-based diets. While we did not directly determine PDCAAS or DIAAS for our specific formulation, literature-based values support the completeness of the soy–oat protein blend [17,19].

**Materials:** Soy flakes were purchased from Soy Austria GmbH (Wien, Austria). Distilled water (catalogue code W3513), citric acid (catalogue code W230618, ≥99.5%, molecular weight: 192.12), and standard laboratory equipment (blender, vacuum filtration unit, pH meter, refrigerated centrifuge) were used for protein extraction.


**Extraction steps:**


-*Hydration*:

A batch of 100 g of soy flakes was weighed and submerged in three times their weight of distilled water (1:3 *w*/*w* ratio) and left to soak overnight (~16 h) at ambient temperature (~20–22 °C). Hydration facilitates cell wall softening and improves subsequent protein extraction efficiency by enhancing cell rupture. Although this method successfully produced a protein-rich extract, the exact extraction yield, defined as the mass of recovered protein relative to the initial soy flake mass, was not quantified in this preliminary study. Specifically, from an initial 100 g batch of soy flakes, we obtained approximately 26 g of recovered protein material, resulting in an extraction yield of ~26%. This value was calculated based on the dry weight of the final protein isolate relative to the initial soy flake mass, following the precipitation and centrifugation steps.

-*Blending*:

The hydrated soy flakes were drained and homogenized using a laboratory blender (8000 rpm) until a uniform slurry was achieved. Homogenization mechanically disrupted cellular structures, promoting the release of soluble proteins into the aqueous medium.

-*Filtration*:

The slurry was subjected to vacuum-assisted filtration through a Buchner funnel equipped with Whatman No.1 filter paper (Buckinghamshire, UK). The filtrate, a crude protein solution, was collected separately, while solid residues (cellular debris) were discarded.

-*Protein precipitation*:

The pH of the crude extract was gradually lowered to the isoelectric point of soy proteins (~pH 4.5–4.8), a range selected to accommodate natural variability in protein subtypes and ensure complete precipitation. The careful addition of citric acid under constant stirring, monitored using a calibrated digital pH meter, ensured precise control of the process. Acid-induced precipitation promotes the aggregation of protein molecules, thereby facilitating their separation from soluble carbohydrates and other minor components.

-*Separation and storage*:

The precipitated protein was allowed to settle for 30 min at 4 °C, enhancing precipitation completeness. Subsequently, the mixture was centrifuged at 5000 rpm for 10 min using a Thermo Scientific™ Sorvall™ ST 16 R centrifuge (Agilent Technologies, Santa Clara, CA, USA) equipped with a TX-400 swinging bucket rotor to pellet the protein fraction. The supernatant was discarded, and the protein pellet was gently resuspended in minimal cold distilled water and stored at 4 °C for further biochemical and functional analyses.

### 2.2. Methods of Biochemical Analysis of Protein Extract

The protein content and chemical structure of the soy protein extract were evaluated through a combination of qualitative and spectroscopic methods, ensuring both verification of protein presence and characterization of the molecular components.

The qualitative presence of proteins in the soy extract was assessed using the Biuret method, a classical and widely employed assay for detecting peptide bonds in protein molecules [20]. The Biuret method used in this study followed standard procedures without modification, serving as a qualitative tool to confirm protein presence in the extract. The principle of the Biuret reaction involves the formation of a violet-colored complex when proteins, containing two or more peptide bonds, react with copper(II) ions under alkaline conditions. The method derives its name from the compound *biuret* (produced by heating urea), which exhibits a similar chemical reactivity toward copper salts.

#### 2.2.1. Preparation of Biuret Reagent

All reagents were weighed precisely using an analytical balance. In a Berzelius beaker, 0.3 g of copper sulfate (CuSO_4_·5H_2_O) and 1.2 g of sodium potassium tartrate (Rochelle salt) were dissolved in 50 mL of distilled water, measured using a 50 mL graduated cylinder. Subsequently, 0.2 g of potassium iodide (KI) was added as a chelating agent to stabilize the copper ions and prevent their reduction during reagent storage. The solution was transferred into a 100 mL volumetric flask. Then, 2 M sodium hydroxide (NaOH) solution was added to provide the necessary alkaline medium, and distilled water was added to bring the final volume to 100 mL. The mixture was homogenized thoroughly to ensure complete dissolution of all salts.

#### 2.2.2. Assay Procedure

Equal volumes of freshly prepared Biuret reagent and protein extract were mixed in clean test tubes. The reaction mixture was incubated at room temperature for 15 min. The appearance of a violet or purple coloration confirmed the presence of peptide bonds, thereby verifying the presence of proteins in the soy extract.

#### 2.2.3. Quantitative and Structural Characterization

To further characterize the chemical composition and confirm the presence of protein-specific molecular features, Ultraviolet-Visible (UV-Vis) Spectroscopy (Shimadzu Pharma Spec 1700 UV-Vis spectrophotometer, San Diego, CA, USA) and Fourier Transform Infrared (FTIR) Spectroscopy (Vertex 70 spectrophotometer, Bruker, Germany) were employed. The absorbance spectrum was recorded over the range of 200–400 nm. A characteristic absorption peak observed at approximately 256 nm indicated the presence of aromatic amino acids such as tryptophan and tyrosine. While this is slightly lower than the typical 280 nm peak expected for aromatic amino acids, similar shifts have been reported in plant-derived protein extracts where sample matrix complexity or the presence of secondary metabolites (e.g., polyphenols or flavonoids) can influence absorbance characteristics. This still supports the presence of proteinaceous compounds in the extract, though it may reflect overlapping absorbance from non-protein components.

Spectra were recorded in the range of 4000–400 cm^−1^ using the attenuated total reflectance (ATR) mode with 32 scans per sample at a resolution of 4 cm^−1^. All measurements were performed in collaboration with the Department of Chemistry at the Politehnica University of Timișoara. Prior to analysis, the samples were oven-dried at 60 °C for 2 h to eliminate moisture, which could interfere with absorbance in the 3200–3600 cm region.

FTIR analysis was conducted to confirm the presence of functional groups characteristic of protein molecules. The dried soy protein extract was finely ground and analyzed using a Bruker Vertex 70 FTIR (Billerica, MA, USA) spectrometer. Spectra were recorded in the range of 4000–400 cm^−1^ using the attenuated total reflectance (ATR) mode with 32 scans per sample at a resolution of 4 cm^−1^. All measurements were performed in collaboration with the Department of Chemistry at the Politehnica University of Timișoara. Prior to analysis, the samples were oven-dried at 60 °C for 2 h to eliminate moisture, which could interfere with absorbance in the 3200–3600 cm^−1^ region.

### 2.3. Selection of Ingredients and Protein Bar Formulation

The formulation of the experimental protein bars began with a systematic selection of ingredients aimed at achieving a nutritionally balanced, stable, and consumer-acceptable product. To inform our selection process, we conducted a comprehensive market analysis by reviewing the nutritional composition of commercially available protein bars through various online retailers and product databases. This survey provided insights into common ingredient choices, typical macronutrient distributions, and market trends in protein bar formulations.

Based on this review, we identified key ingredients that would collectively support a balanced macronutrient profile, ensuring adequate proportions of proteins, carbohydrates, fibers, and fats. The nutritional goals for the final product were aligned with the general recommendations for functional snacks targeting active individuals, as well as the macronutrient intake guidelines established by the European Food Safety Authority (EFSA, 2017) [21].

In addition to nutritional adequacy, ingredient selection was guided by functional considerations critical for product quality. Ingredients were chosen to maintain bar cohesiveness and prevent excessive hardening or crumbling during storage. Natural flavoring agents were prioritized to enhance palatability without reliance on synthetic additives. Components with inherent preservative effects (such as fiber- and antioxidant-rich ingredients) were preferred to extend shelf life without the inclusion of artificial preservatives. Ingredient affordability was evaluated to ensure that the final product would be accessible to a broader consumer base without compromising its nutritional or sensory properties.

Following the selection of ingredients, we determined their respective quantities for formulation. This process involved referencing standardized recipes for high-protein bars commonly found in the scientific and industry literature; considering the average caloric values and macronutrient profiles observed in leading commercial protein bars; aligning macronutrient content per serving with EFSA recommendations for energy and nutrient intake, particularly for individuals engaged in moderate to high levels of physical activity.

The final formulation was thus rationalized to deliver a balanced intake of proteins (≥20% of total weight), complex carbohydrates, dietary fiber, and healthy fats, providing a nutritionally dense and functional snack without resorting to synthetic stabilizers, added sugars, or artificial flavorings.

The final protein bar formulation incorporated carefully selected ingredients chosen for their complementary nutritional profiles and functional properties. The ingredients included soy protein extract, oatmeal, dates, chia seeds, peanut butter, and cinnamon.

The ingredients were weighed precisely using an analytical balance according to the following formulation: 100 g soy protein extract, 150 g oatmeal, 60 g peanut butter, 40 g dates, 100 g chia seeds, and 4 g cinnamon powder (approximately one tablespoon). The proportional contribution of each ingredient to the final bar formulation is illustrated in Figure 1.


**Preparation procedure**


-*Hydration of fiber sources*:

Chia seeds and dates were placed in separate containers and submerged in sufficient distilled water to fully cover their surfaces. The chia seeds were allowed to swell and form a gel-like consistency, while the dates softened, facilitating subsequent blending.

-*Preparation of binding paste*:

After soaking, the dates were thoroughly drained to remove excess moisture. The drained dates were then combined with peanut butter and cinnamon and processed in a high-speed blender until a homogeneous, sticky paste was obtained. This mixture acted as the primary binding matrix for the bar.

-*Incorporation of dry ingredients*:

The hydrated chia seeds and dry oatmeal were gradually added to the binding paste. Mixing was performed manually or using a stand mixer until a cohesive, dough-like consistency was achieved, ensuring even distribution of all components throughout the mixture.

-*Shaping and setting*:

The protein bar dough was transferred into a rectangular tray lined with parchment paper and was spread uniformly using a spatula or rolling pin to achieve a consistent thickness across the entire surface. Cold processing techniques were employed, deliberately avoiding baking or other heat treatments to preserve bioactive compounds and prevent Maillard reactions, non-enzymatic browning processes that typically occur at high temperatures between reducing sugars and amino acids. While this approach limits thermally induced nutrient degradation and undesirable flavor development, we acknowledge that enzymatic or oxidative changes may still occur during refrigerated storage, particularly in the absence of synthetic preservatives [5].

-*Refrigeration and portioning*:

The tray was covered and refrigerated at 4 °C for a minimum of 2 h. The cold setting allowed the mixture to firm up by stabilizing the water–fat–protein matrix through physical entanglement and hydration forces. Once adequately firm, the slab was removed from refrigeration and cut manually into individual portions, each weighing approximately 50 g.

-*Packaging and storage*:

Each protein bar portion was individually wrapped in aluminum foil to protect the product from oxidative degradation, particularly of the lipid components. The bars were stored at 4 °C to maintain freshness and inhibit microbial growth during the evaluation period.

This method of cold processing preserves the nutritional and sensory properties of the ingredients, preventing the thermal degradation of vitamins, phytochemicals, and sensitive amino acids [5]. Additionally, it avoids the development of undesirable flavors or potentially harmful Maillard reaction products that can form during high-temperature processing.

The protein bars were stored in a refrigerator at 4 °C to maintain their freshness and prevent spoilage. They were kept at this temperature for up to one week, ensuring their stability and nutritional value during storage. The final formulation of the protein bar, including the ingredient composition and quantities, is provided in Table 1.

### 2.4. Selection of a Commercial Protein Bar for Benchmarking

For benchmarking purposes, a commercial protein bar was selected and purchased from a local supermarket. The chosen product contained soy protein as its primary protein source, aligning with the protein base utilized in our experimental bar formulation. This ensured that the comparison remained relevant and scientifically justified, minimizing variability due to differences in protein origin and type. The comparative evaluation was designed to comprehensively assess not only the nutritional adequacy and sensory quality of the two protein bars but also their practical affordability and alignment with health-conscious consumer preferences.

The comparative analysis between the experimental and commercial protein bars focused on several key parameters: nutritional composition, shelf-life assessment, ingredient quality, and cost-effectiveness.

Both products were evaluated for their macronutrient profiles, including protein, total carbohydrates, fiber content, total fats, and overall caloric value. Nutritional information for the commercial bar was sourced from the product label, and its full ingredient list is provided in Appendix A to enable direct comparison with the experimental formulation. The composition of the experimental bar was calculated based on the known quantities and nutritional values of its ingredients. This analysis aimed to determine differences in nutrient density, the proportion of functional macronutrients, and the degree to which each bar supported health-oriented nutritional goals, such as high protein and high fiber content with reduced added sugars.

Shelf life was evaluated by storing both bars under identical refrigerated conditions (4 °C) and monitoring changes in physical properties, including texture, flavor, aroma, and appearance, over a defined period. Particular attention was given to the role of preservatives: the commercial bar, containing additives such as lecithins and other emulsifiers, was expected to exhibit an extended shelf life compared to the experimental bar, which relied solely on natural ingredients without artificial preservatives. The potential influence of the processing method, cold processing for the experimental bar versus possible heat treatments for the commercial product, was also considered in evaluating shelf stability.

A qualitative assessment of the ingredient lists was conducted. This involved examining the source and degree of processing of the ingredients, identifying the presence of natural versus synthetic components, and noting the use of additives such as artificial sweeteners, stabilizers, hydrogenated oils, and emulsifiers. The goal was to assess the overall health impact and naturalness of each formulation, emphasizing clean-label attributes in the experimental bar.

Economic analysis compared the total cost of ingredients used in the homemade protein bar to the retail price of the commercial bar. Ingredient costs for the experimental bar were calculated based on market prices for raw materials, while the commercial bar’s price was documented at the point of purchase. This analysis sought to determine whether the homemade formulation provided a more affordable, nutritionally rich, and additive-free alternative to its commercial counterpart.

### 2.5. Comparative Evaluation Methods

A soy-based commercial protein bar was selected from a local retailer for benchmarking against the experimental formulation. To assess the nutritional and functional quality of the experimental protein bar, a comparison was made with a commercially available product using the following criteria: nutritional composition, shelf stability, ingredient quality, and cost-effectiveness.

Nutrient data for the commercial product were obtained directly from the manufacturer’s label and technical leaflet. For the experimental bar, macronutrient content was calculated based on the known composition and the nutritional information provided by suppliers and standard food composition databases.

Both bars were stored at 4 °C for up to 14 days. Visual appearance, aroma, and texture were assessed on days 0, 7, and 14. Firmness was evaluated manually by compressing the samples and recording qualitative observations. No instrumental texture or microbiological analyses were performed in this pilot study.

The cost of ingredients for the experimental bar was calculated using average local supermarket prices. The commercial bar’s cost per 100 g was recorded at the point of purchase. All prices were standardized to RON per 100 g for direct comparison.

## 3. Results and Discussions

In recent years, the popularity of high-protein bars has increased considerably, particularly among athletes and fitness enthusiasts. These products offer a convenient, nutrient-dense option that supports muscle growth and recovery [4]. Typically formulated with a balanced mix of protein (20–40%), carbohydrates (10–50%), and fats (10–15%) [8], protein bars aim to deliver concentrated nutrients while preventing deficiencies and promoting efficient absorption. Carbohydrates often come from glucose, fructose, and maltose syrups, which provide energy and act as binding agents, while additives like flavor enhancers and stabilizers improve texture, shelf life, and taste [5].

Despite their convenience, some commercial protein bars raise concerns due to high added sugar content, artificial preservatives, and low-quality protein sources, potentially diminishing their intended health benefits [21,22,23,24,25].

However, not all added ingredients are beneficial. Some commercial bars still include hydrogenated oils, a major source of trans fats, which are linked to elevated LDL cholesterol, reduced HDL cholesterol, and increased cardiovascular risk [26]. Moreover, food additives, though improving flavor, texture, and shelf life, may pose health risks for sensitive individuals, potentially causing allergic reactions or digestive disturbances [27]. As a result, ingredient selection remains crucial to maximize the health benefits and safety of protein bars.

The ingredients used in our protein bar formulation included soy protein extract, oatmeal, dates, chia seeds, peanut butter, and cinnamon, each contributing significantly to the overall nutritional and functional profile:-Soy protein extract served as the primary protein source, providing a complete amino acid profile essential for supporting muscle synthesis and metabolic health.-Oatmeal and chia seeds were selected for their high content of soluble dietary fiber, promoting gastrointestinal health, improving glycemic control, and contributing to the final texture and structural stability of the bars.-Dates performed a dual function: acting as a natural sweetener by providing intrinsic sugars and serving as an effective natural binding agent due to their viscous texture, helping to hold the ingredients together without the need for synthetic binders.-Peanut butter was incorporated as a source of healthy monounsaturated and polyunsaturated fatty acids, contributing both to the creamy mouthfeel of the bars and to the overall energy density required for a functional snack.-Cinnamon was added to enhance flavor naturally and to potentially provide antioxidant benefits.

### 3.1. Biochemical Analysis of Protein Extract

The Biuret test confirmed the presence of protein in the extracted soy material. This qualitative result indicated successful precipitation and recovery of peptide-bond-containing molecules.

Despite its widespread use, the Biuret method has several known limitations regarding specificity. Notably, the reaction is not exclusively selective for peptide bonds; it can also yield positive results in the presence of non-protein compounds containing two carbonyl groups (C=O) linked by a nitrogen or carbon atom. Such compounds can interfere with the assay by producing similar colorimetric changes, thereby introducing potential sources of error when interpreting results, especially in complex mixtures where non-protein nitrogenous substances may be present.

Furthermore, while the Biuret assay is effective for estimating overall protein or polypeptide content, it does not provide information on the specific amino acid composition or the molecular characteristics of the proteins detected. It cannot differentiate between different types of proteins or peptides, nor can it quantify individual amino acids. The method, therefore, serves primarily as a qualitative or semi-quantitative tool, offering an indirect assessment based on the aggregate peptide bond concentration.

Nevertheless, the Biuret method remains a reliable and straightforward technique for the initial screening of protein presence in biological samples. Its ease of implementation, relatively low cost, and ability to deliver rapid results make it a valuable tool in preliminary biochemical analyses. For more detailed or specific characterization of proteins, complementary analytical techniques, such as amino acid profiling, SDS-PAGE electrophoresis, or mass spectrometry, would be required.

Biochemical analysis utilizing UV-Vis spectroscopy provides critical insights into the presence and behavior of light-absorbing compounds within a sample. This technique operates on the principle that molecules absorb specific wavelengths of ultraviolet or visible light, leading to electronic transitions within their molecular structures. By measuring the absorbance at different wavelengths, UV-Vis spectroscopy enables the identification, qualitative assessment, and, under calibrated conditions, the quantification of specific substances based on their characteristic absorption profiles. In the context of protein analysis, UV-Vis spectroscopy is particularly valuable for detecting aromatic amino acids, such as tryptophan, tyrosine, and phenylalanine, which exhibit strong absorption in the ultraviolet region due to the presence of conjugated π-electron systems within their aromatic rings. These structural features make aromatic residues important chromophores that can be exploited for protein detection and characterization.

During our analysis, the absorbance spectrum of the soy protein extract was recorded across the wavelength range of 200–400 nm. As illustrated in Figure 2, a distinct absorption peak was observed at approximately 256 nm. This peak is characteristic of the π → π* electronic transitions associated with the aromatic rings of amino acids such as tryptophan and tyrosine. The presence of this absorption band indicates that the extracted protein contains these aromatic residues, providing qualitative confirmation of the proteinaceous nature of the sample.

The detection of aromatic amino acids is significant because they contribute not only to the structure and stability of proteins but also to their biochemical reactivity and interactions with other molecules. Furthermore, the specific absorbance at 256 nm supports the presence of intact peptide structures within the sample, suggesting that the extraction and precipitation procedures preserved key molecular features necessary for functional protein applications. UV-Vis spectroscopy offers several advantages in protein studies: it is non-destructive, preserving the sample for further analysis; it is rapid, enabling real-time or high-throughput screening; it provides a qualitative and semi-quantitative means of assessing protein concentration and purity, particularly when combined with established calibration curves. The standard protein absorbance peak typically centers around 280 nm, due to the presence of aromatic residues such as tryptophan and tyrosine. However, as noted in a recent study, proteins also exhibit distinct absorbance maxima at 275 nm (tyrosine) and 258 nm (phenylalanine), and absorbance at these lower wavelengths can provide meaningful insight into protein structure and conformation. The authors of a study demonstrated that absorbance ratios at 280, 275, and 258 nm can serve as a sensitive probe for protein foldedness and molecular interactions [28]. In our case, the observed peak at approximately 256 nm is consistent with the absorbance behavior of aromatic residues, particularly phenylalanine, and may also reflect the specific conformational or environmental conditions of the soy protein extract. Therefore, while not centered at 280 nm, the peak observed supports the presence of intact aromatic amino acids and folded protein structures, aligning with current literature on UV absorbance in protein characterization. Overall, the presence of a strong absorption peak at 256 nm validates the success of the protein extraction process and sets the foundation for further functional or structural analyses of the formulated protein products.

FTIR spectroscopy is a highly valuable analytical technique for the characterization of proteins and other organic compounds, offering detailed insights into molecular structures and the identification of functional groups. FTIR operates on the principle that molecular bonds absorb infrared radiation at characteristic frequencies corresponding to specific vibrational modes. These vibrations, including stretching and bending movements of chemical bonds, result in distinct absorption bands within the infrared spectrum. The pattern and intensity of these absorption bands provide crucial information regarding the chemical composition, bond types, and molecular environment of the analyzed sample.

In protein analysis, FTIR spectroscopy is particularly effective due to its sensitivity to polar functional groups, such as amides, hydroxyls, and amines, which are key components of protein secondary and tertiary structures. Moreover, FTIR allows for the non-destructive evaluation of samples, making it suitable for food science applications where maintaining sample integrity is important. To ensure accurate and interference-free FTIR measurements, the soy flakes were carefully oven-dried at a controlled temperature prior to analysis. This step was crucial to eliminate residual moisture, as water strongly absorbs infrared radiation in the 3000–3600 cm^−1^ region, potentially obscuring relevant protein absorption bands.

The FTIR spectrum of the dried soy flakes (Figure 3) revealed several characteristic absorption bands, each corresponding to specific functional groups present in the sample:-A broad, intense band was observed in the region between 3500 and 3200 cm^−1^, attributed to O–H stretching vibrations from hydroxyl groups and N–H stretching vibrations from amine groups. These functional groups are abundant in proteins, polysaccharides, and other biomolecules, indicating the presence of hydrophilic structures within the soy matrix.-A prominent absorption peak near 1650 cm^−1^ corresponds to C=O stretching vibrations of amide linkages, known as the Amide I band. This band is a hallmark of peptide bonds and provides strong evidence for the presence of proteins in the sample. The Amide I region is often used to assess protein secondary structures, including α-helices and β-sheets.-Absorption bands around 1400 cm^−1^ were assigned to the bending vibrations of aliphatic C–H groups. These signals reflect the presence of nonpolar hydrocarbon chains, likely originating from fatty acid residues or aliphatic side chains of amino acids embedded within the soy proteins or associated lipids.-Additional medium-intensity bands detected near 1100 cm^−1^ are associated with C–O and C–N stretching vibrations. These bands are characteristic of glycosidic linkages in carbohydrates and peptide bonds in proteins, further confirming the composite nature of the soy material, consisting of both protein and polysaccharide constituents.

The identification of these key absorption bands provides a comprehensive molecular fingerprint of the soy flakes, confirming the successful retention of major functional groups relevant to protein structure. FTIR spectroscopy thus not only verifies the presence of proteins but also offers valuable structural information about the interactions between proteins, carbohydrates, and lipid components in the food matrix.

By applying FTIR spectroscopy, we gain critical insights into the functional properties of soy-based ingredients, such as their hydration potential, emulsification capabilities, and stability—features essential for their effective use in food product formulation, particularly in applications such as high-protein nutritional bars.

### 3.2. Comparative Analysis of the Experimental Protein Bar and a Commercial Counterpart

#### 3.2.1. Nutritional Value of Protein Bars

The nutritional composition of the experimental bar was determined using triplicate measurements, and mean ± standard deviation values have been provided in Table 2. The comparative nutritional analysis highlights the advantages of the experimental protein bar, particularly its significantly higher fiber content (13.46 g/100 g vs. 3.7 g/100 g) and increased protein concentration (25.52 g/100 g vs. 18.5 g/100 g) compared to the commercial product. These differences underline the nutritional optimization achieved in the experimental formulation, emphasizing its design focus on delivering a more balanced and nutrient-dense profile suitable for health-conscious consumers.

According to the European Union regulations on nutrition claims (European Commission, https://food.ec.europa.eu/) [29], both products qualify for the label “source of protein,” as their protein content exceeds 12% of the total energy value. However, the experimental protein bar meets the more stringent classification of “high-protein,” which requires that at least 20% of the product’s energy is derived from protein. This distinction positions the experimental bar as a superior option for individuals seeking enhanced protein intake, such as athletes, physically active individuals, or those following specialized dietary regimens.

Moreover, the experimental protein bar also fulfills the regulatory criteria for a “high-fiber” product, containing no less than 6 g of fiber per 100 g of product, as stipulated by European Commission guidelines. High dietary fiber intake is associated with multiple health benefits, including improved gastrointestinal health, better glycemic control, and enhanced satiety, which collectively contribute to better overall nutritional outcomes.

In addition to its macronutrient advantages, the experimental protein bar qualifies for the claim of “does not contain refined sugars or artificial sweeteners”, further enhancing its appeal as a healthy snack alternative. The experimental bar contains no added sugars, relying instead on dates as a natural sweetener and binder. This feature is particularly relevant given growing consumer demand for low-sugar, additive-free products. In contrast, the commercial product includes glucose syrups and artificial additives, potentially undermining its health appeal. This characteristic is particularly relevant given the increasing consumer demand for products with reduced sugar content, motivated by public health recommendations aimed at lowering the incidence of metabolic disorders such as obesity and type 2 diabetes.

It is important to note that the nutritional data for the commercial protein bar were obtained directly from the manufacturer’s product labeling and official leaflets, ensuring the accuracy and standardization of the comparative analysis. By relying on verified nutritional information, the integrity of the comparison is preserved, allowing for a fair and transparent evaluation of the two products.

Statistical analysis (mean ± SD) was performed for the experimental protein bar based on triplicate measurements. However, direct statistical comparison with the commercial protein bar was not possible, as the nutritional values for the commercial product were sourced exclusively from its manufacturer-provided label, without available standard deviations or replicates. Therefore, no *p*-values or significance tests could be validly applied between the two products.

#### 3.2.2. Shelf Life of Protein Bars

One notable limitation of the experimental protein bar is its relatively short shelf life, which extends to approximately seven days when stored under refrigerated conditions (4 °C). In contrast, the commercial protein bar maintains its stability and sensory quality for up to 90 days under similar conditions. This significant difference highlights the critical role that additives and preservatives play in enhancing product longevity and maintaining quality over extended storage periods. While the absence of preservatives contributed to a reduced shelf life (7 days vs. 90 days), this outcome aligns with the formulation’s clean-label goals. However, it also emphasizes the need for further work on natural preservation methods or improved packaging solutions to enhance stability without compromising ingredient integrity.

However, microbial testing (e.g., total plate count) was not performed during this study. The shelf-life assessment relied on qualitative observations of texture, aroma, and visual appearance under refrigerated conditions (4 °C). While these parameters provide useful preliminary insights into product stability, they do not substitute for objective microbiological testing. We recognize this as a limitation and plan to include microbiological assessments in future research to ensure a comprehensive evaluation of microbial safety and shelf-life stability.

The prolonged shelf life of the commercial protein bar can be largely attributed to the incorporation of sunflower lecithin in its formulation. Lecithins are complex mixtures predominantly composed of phospholipids, which are insoluble in acetone and often accompanied by minor components such as triglycerides, fatty acids, and carbohydrates. Structurally, lecithins possess amphiphilic characteristics, featuring both hydrophilic (polar, water-attracting) and lipophilic (non-polar, fat-attracting) domains. This dual affinity enables lecithins to act effectively at the oil–water interface, stabilizing emulsions formed during homogenization processes [30].

The emulsifying action of lecithins is crucial in maintaining product consistency, preventing phase separation, and preserving the homogeneity of multi-component systems such as protein bars. By forming stable interfacial films around fat droplets or dispersed phases, lecithins help protect the product against mechanical and chemical destabilization. Beyond their primary role as emulsifiers, lecithins also contribute significantly to oxidative stability. They act as physical barriers against oxygen diffusion and may possess intrinsic antioxidant properties that slow down the oxidative degradation of sensitive lipids and other ingredients, as supported by different studies, which demonstrated the emulsifying and oxidative stability functions of modified sunflower lecithins [30,31,32,33]. This protection preserves the texture, flavor, color, and nutritional integrity of the product over time.

In addition to formulation factors, external variables, such as packaging techniques, storage conditions, and manufacturing processes, are also pivotal in extending shelf life. Effective packaging materials minimize the exposure of the product to oxygen, moisture, and light, all of which can accelerate spoilage [34]. Controlled manufacturing environments and strict cold chain maintenance further reduce the risks of microbial contamination and ingredient degradation. In contrast, the absence of synthetic preservatives in the experimental protein bar, while aligning with clean-label and natural product trends, inherently limits its stability. Without emulsifiers or antioxidant additives, natural ingredients are more vulnerable to moisture absorption, lipid oxidation, microbial growth, and textural deterioration, necessitating prompt consumption within a short timeframe.

The limited shelf-life of the experimental protein bar appears to be influenced primarily by the overall water activity and moisture content, which can promote microbial growth and lipid oxidation over time.

While protein composition can have an impact on stability, our data suggest that water activity plays a more significant role in this case. Regarding soy lecithin, we acknowledge that it is commonly used as an emulsifier and can contribute to improving texture and shelf life by stabilizing fat–water interfaces and potentially reducing oxidation. However, in our initial formulation, soy lecithin was not included as our focus was on optimizing the core nutritional and sensory properties first.

Moving forward, we recognize that incorporating emulsifiers like soy lecithin could enhance shelf-life stability, and we plan to explore this in subsequent formulations to address these concerns more effectively.

The limited shelf-life of the experimental protein bar can be attributed in part to its relatively high water activity (a∨w) and near-neutral pH, which, together, create favorable conditions for microbial growth and spoilage. Although a∨w and pH were not directly measured in this pilot study, the use of high-moisture ingredients such as dates and chia gel likely resulted in a water activity value above 0.85—a known threshold for bacterial and fungal proliferation in food products. The absence of preservatives further compounded the susceptibility to microbial degradation.

To address this limitation, future research will include precise measurement of a˅w and pH, followed by targeted interventions. One potential approach involves incorporating natural antimicrobial agents (e.g., citric acid, essential oils, or plant-based extracts such as rosemary or clove) that can inhibit microbial growth without compromising clean-label goals. Additionally, applying modified atmosphere packaging, vacuum sealing, or moisture-barrier films could help extend shelf stability by reducing oxygen exposure and controlling moisture migration. These strategies will be considered in future reformulations aimed at enhancing microbial safety while maintaining the natural composition of the product.

#### 3.2.3. Qualitative Assessment of Ingredients and Their Nutritional Relevance

In both the experimental and commercial protein bars, soy protein serves as the primary protein source. However, in the commercial product, soy protein is combined with additional ingredients, notably tapioca starch and salt, which together contribute approximately 16% of the final product’s composition. Although this figure represents the combined contribution of these ingredients, the specific proportion of soy protein alone is not disclosed by the manufacturer, limiting the precision of direct protein content comparisons between the two products.

When examining fat sources, distinct differences emerge between the two bars. In the commercial protein bar, a substantial portion of the fat content is derived from palm oil, an ingredient that has been the subject of extensive debate within nutritional science. Some studies have identified potential therapeutic benefits of palm oil, including its capacity to increase antioxidant levels [35], lower the risk of atherosclerosis, and favorably modulate lipid profiles by elevating high-density lipoprotein (HDL) cholesterol while reducing low-density lipoprotein (LDL) cholesterol [36]. These findings suggest that, under controlled consumption, palm oil could contribute positively to cardiovascular health.

However, contrasting research raises important concerns about palm oil consumption, primarily due to its high content of saturated fatty acids. Excessive intake of saturated fats has been consistently associated with an elevated risk of cardiovascular diseases, including coronary artery disease and stroke [35,36]. Thus, while palm oil may offer some health benefits in moderation, its regular and high consumption could pose significant health risks, especially within populations already vulnerable to cardiovascular conditions.

Conversely, the experimental protein bar incorporates peanut butter as its principal fat source. Peanut butter is rich in unsaturated fatty acids, particularly monounsaturated fats, which are widely recognized for their cardioprotective properties. Numerous studies have associated the regular consumption of unsaturated fats with improvements in lipid profiles, including reductions in LDL cholesterol and increases in HDL cholesterol, ultimately contributing to a decreased risk of cardiovascular diseases [37]. Furthermore, peanut butter provides additional nutritional benefits, including a good supply of plant-based protein, fiber, vitamins (such as vitamin E), and minerals (such as magnesium and potassium).

By selecting peanut butter as the primary lipid component, the experimental protein bar emphasizes a health-oriented approach, favoring nutrient-dense, minimally processed ingredients known to support long-term cardiovascular health. This choice offers a more favorable fat profile compared to the commercial product relying on palm oil, aligning better with contemporary nutritional guidelines that advocate for the replacement of saturated fats with unsaturated fats in the diet.

Although both protein bars utilize soy as their core protein source, the experimental formulation demonstrates a more transparent and health-conscious selection of ingredients, particularly in its use of unsaturated fat-rich peanut butter instead of saturated fat-rich palm oil. This strategic choice not only enhances the nutritional profile of the product but also aligns with current dietary recommendations aimed at improving public health outcomes.

Both the experimental and commercial protein bars utilize soy protein as their primary protein source. However, the formulation of the commercial bar differs in that the soy protein is combined with additional ingredients, such as tapioca starch and salt. Together, these components contribute approximately 16% to the overall mass of the final product. It is important to note that this percentage reflects the combined contribution of all listed ingredients, and the specific proportion of soy protein alone is not explicitly stated by the manufacturer. Consequently, the actual concentration of soy protein in the commercial bar remains unclear, making direct nutritional comparisons somewhat limited in precision.

A key difference between the two bars also lies in the type of fat sources utilized. In the commercial protein bar, a significant portion of the fat content is derived from palm oil, an ingredient that has generated considerable debate in nutrition science. Some research highlights the potential therapeutic effects of palm oil, including its ability to enhance antioxidant defenses [35], reduce the progression of atherosclerosis, and improve lipid profiles by increasing high-density lipoprotein (HDL) cholesterol while lowering low-density lipoprotein (LDL) cholesterol [36]. These benefits suggest that, when consumed in moderation, palm oil may exert positive effects on cardiovascular health.

However, other studies raise concerns regarding palm oil’s health implications. Palm oil is notably high in saturated fatty acids, and excessive intake of saturated fats has been strongly associated with an increased risk of cardiovascular diseases, including coronary artery disease and stroke [35,36]. The high saturated fat content of palm oil makes it a controversial choice, especially in populations where overall saturated fat consumption is already elevated. While palm oil may offer some protective benefits at moderate intake levels, its inclusion in processed foods warrants cautious evaluation.

In contrast, the experimental protein bar uses peanut butter as its principal lipid source. Peanut butter is recognized for its high content of unsaturated fatty acids, particularly monounsaturated fats, which are known for their protective effects against cardiovascular disease [38]. However, it is important to note that peanuts are susceptible to contamination with aflatoxins—mycotoxins produced by *Aspergillus* species—which have been associated with carcinogenic and hepatotoxic effects [39,40]. While regulatory limits and quality control measures (e.g., sorting, proper storage, and roasting) significantly reduce aflatoxin levels in commercially available peanut products, this remains an important consideration in evaluating their overall health impact [41,42]. Regular consumption of unsaturated fats has been linked to favorable lipid profile modulation, including reductions in LDL cholesterol levels and improvements in HDL cholesterol concentrations [37]. In addition to its beneficial fat composition, peanut butter provides valuable nutrients such as plant-based protein, dietary fiber, and essential micronutrients like magnesium and potassium, all of which contribute to its reputation as a heart-healthy food.

By incorporating peanut butter, the experimental protein bar focuses on nutrient-dense, minimally processed ingredients that promote long-term cardiovascular health. The use of unsaturated fats aligns with current dietary guidelines that advocate for reducing saturated fat intake in favor of healthier fat sources. This strategic choice results in a more favorable nutritional profile for the experimental bar compared to the commercial alternative that relies heavily on palm oil.

Overall, while both protein bars are anchored by soy protein, the experimental bar demonstrates a clearer commitment to health-oriented formulation, offering advantages not only in protein quality but also in fat composition. This formulation approach provides consumers with a cleaner, nutritionally superior option that supports cardiovascular health and broader wellness objectives.

While the health effects of soy protein have raised concerns, particularly in relation to hormone-related cancers, recent comprehensive reviews have clarified that these concerns are mainly associated with high-dose isoflavone supplements rather than whole soy foods [43]. As noted by Messina et al., moderate consumption of traditional soy products and soy-derived protein from food sources is considered safe and may even offer protective benefits, especially for breast cancer survivors [44]. Therefore, incorporating soy protein from whole food sources into protein bar formulations aligns with current scientific evidence supporting its safety and potential health benefits.

Discussions surrounding the improvement of protein bars often emphasize enhancing the benefits of antioxidants, exogenous compounds that not only help protect the cardiovascular system but also support a variety of endogenous processes within the body. While these protein bars may have a shorter shelf life compared to conventional commercially available options, the inclusion of such beneficial ingredients can significantly elevate their nutritional value and overall health impact.

#### 3.2.4. Cost Analysis

Cost analysis revealed that the experimental bar costs approximately 25% less per 100 g than the commercial counterpart, reflecting its use of affordable, whole-food ingredients. This makes it a more accessible option for health-conscious consumers.

To evaluate the cost-effectiveness of these bars, we plan to conduct a clinical study aimed at assessing their effects by examining four specific endocrinological parameters. These parameters will be carefully selected for their relevance to the biological processes affected by protein intake, as well as their connection to broader health outcomes, particularly in relation to immune function, endocrine regulation, and cellular homeostasis. The goal of this study is to determine the long-term benefits of incorporating nutritionally balanced protein bars into the diet, with a focus on their biochemical effects on the body.

The development of nutritionally balanced protein bars that prioritize biochemical interactions holds great potential for offering health benefits that go beyond simple nutrient delivery. These bars could improve overall immune system function, support endocrine health, and promote cellular homeostasis, all while reducing the pharmacotoxicity risks commonly associated with mass-produced commercial products. Additionally, they may enhance the bioavailability of essential nutrients, especially when a shorter shelf life encourages quicker consumption and absorption.

A key concern for individuals with endocrinological disorders is the long-term disruption of hormone levels caused by the use of commercially produced protein bars, which may contain artificial preservatives or poorly balanced nutritional profiles. Preventing such endocrine imbalances, and, potentially, mitigating oncogenic processes, requires promoting dietary choices that are both health-enhancing and therapeutically beneficial. By prioritizing natural, antioxidant-rich ingredients, these protein bars aim to support the prevention of chronic diseases, foster holistic health, and minimize the risks associated with the long-term consumption of highly processed foods.

## 4. Conclusions

Protein bars can serve as an effective means to establish nutritional balance and supplement protein intake. Healthy and affordable protein bars can be developed without complex processes, potentially offering greater effectiveness than some commercially available options. By fostering clear communication with consumers and gaining a deep understanding of their needs, the benefits of consuming protein bars can be maximized.

An important factor to consider when evaluating protein bars is the source of protein. One advantage of soy protein over animal-based proteins is its lower cholesterol and saturated fat content, which provides cardiovascular disease prevention benefits. However, the impact of soy protein remains a topic of debate, with some studies highlighting potential drawbacks, including associations with cancer risk. Many health benefits attributed to soy protein are often derived from epidemiological studies showing lower disease incidences in Asian populations with diets rich in soy derivatives. However, it is important to note that these regions also tend to have lower saturated fat intake due to diets high in fiber, fish, whole grains, vegetables, fruits, and legumes beyond soy. To mitigate the potential drawbacks, soy protein can be combined with other beneficial ingredients. Ultimately, the relationship between soy consumption and disease prevention is complex, with studies often offering contrasting interpretations.

The benefits of protein bars are closely linked to the nature, quality, and proportions of ingredients used in their formulation. To substantiate the claims of health benefits, the choice of ingredients must be carefully evaluated. In particular, the presence and role of artificial additives, as well as the type and quantity of sugars and other sweeteners, should be scrutinized. Opting for protein bars that are minimally processed, with natural ingredients and low sugar content, is crucial in avoiding disruptions to biochemical parameters and mitigating common health concerns, particularly cardiovascular issues. Such choices contribute to a balanced diet and a healthier lifestyle.

## Figures and Tables

**Figure 1 foods-14-02141-f001:**
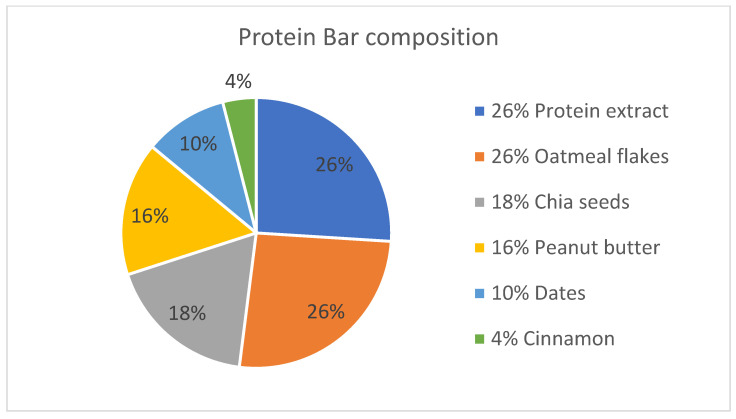
The contribution of each ingredient in obtaining the protein bar.

**Figure 2 foods-14-02141-f002:**
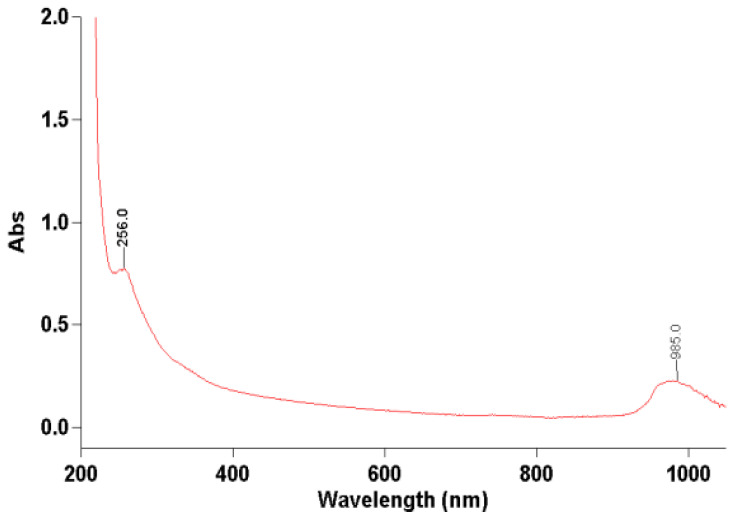
Absorption spectrum of protein extract in ultraviolet spectroscopy.

**Figure 3 foods-14-02141-f003:**
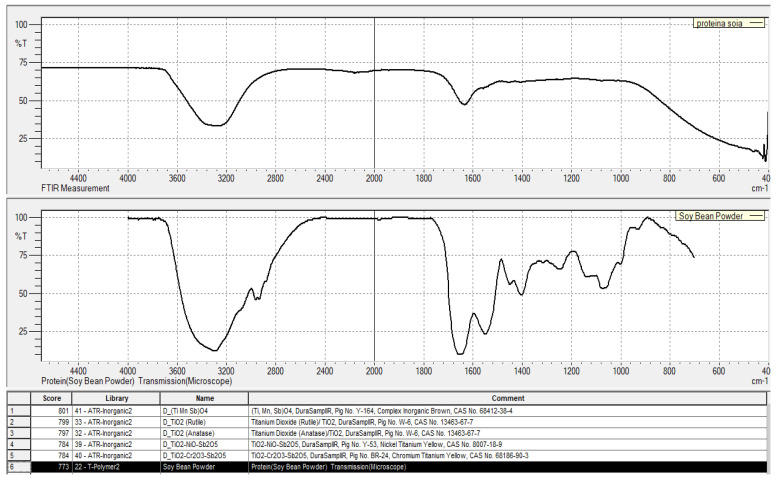
Absorption spectrum of soy flakes in spectroscopy.

**Table 1 foods-14-02141-t001:** Composition of the protein bar per 100 g and per portion (50 g) and the cost of each ingredient (RON—the official currency of Romania).

Ingredient	Content per 100 g (g)	Content per 50 g (g)	Cost per 100 g (RON)
Protein extract from soy flakes	25.97	12.98	2.45
Oatmeal	25.97	12.98	0.99
Peanut butter	15.58	7.79	6.39
Dates	10.38	5.19	2.19
Chia seeds	18.18	9.09	4.30
Cinnamon	3.89	1.94	15.46

Note: The cost of cinnamon (15.46 RON/100 g) reflects retail pricing for ground cinnamon sold in 10 g portions at ~1.55 RON per unit, typical of local consumer packaging.

**Table 2 foods-14-02141-t002:** The nutrient value of the experimental protein bar and the commercial protein bar in 100 g of product and in one serving (50 g).

	Experimental Protein Bar		Commercial Protein Bar	
Macronutrient	100 g	50 g	100 g	50 g
Proteins (g)	25.52 ± 0.34	12.76 ± 0.17	18.5 (label)	9.25 (label)
Carbohydrates (g)	42.41 ± 0.45	21.20 ± 0.23	45.8 (label)	22.9 (label)
Fibre (g)	13.46 ± 0.29	6.73 ± 0.14	3.7 (label)	1.85 (label)
Fats (g)	15.46 ± 0.37	7.73 ± 0.18	20.4 (label)	10.2 (label)

## Data Availability

The original contributions presented in the study are included in the article/Appendix A, further inquiries can be directed to the corresponding author.

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
