# Peer review of "A Comprehensive Study on the Nutritional Profile and Shelf Life of a Custom-Formulated Protein Bar Versus a Market-Standard Product"

_foods, 2025, doi:10.3390/foods14122141_

Round 1
Reviewer 1 Report (New Reviewer)
Comments and Suggestions for Authors
Line 314-350: This information is well-known and not needed in the material and methods section. Please remove it.
Line 409- Any reason for the wide pH range for precipitation?
Line 419-429- Any reason for this text? It should not be part of the material and methods.
Line 464- I believe you have this information in text, so you may not need a table for this information.
Is there any specific reason to make soy protein on your own in the experiment v/s industrial soy protein isolates available with a well-established extraction method? You could have saved time.
Line 518-565- You can trim this down or completely remove this, as the selection factor for ingredients is known. Or you can give this significance in the introduction or use part of the details in the R&D part to discuss your results.
Line 653- What is “RON”?
Line 678-716- This part can be trimmed and written in a few words, “comparison with commercial products considers the following aspects.” There is no need to write a description.
You may need to write methods of how you did the comparison, for example, if you are doing sensory, what sensory methods did you follow? Similarly, for shelf stability, you did an accelerated or normal temperature study, or what instrumental methods were used? That will help to understand what you did in your experiment.
Line 718- 761: This is introduction text rather than R&D information. Please focus on your results and the discussion based on them.
Line 765-773: This is a description or mechanism of the method, which is well known. You can remove it. Also, Remove Figure 2.
Line 788-818- Do you focus on method development or sample analysis? The biuret method is well established for protein quantification. You may need to write in the material method section if you are doing any specific method modification.
Line 890: The material method does not show how you performed FTIR.
Line 890-932: Any relation of this general information to your study? If not, remove it.
Line 959- Any reason for comparing your experimental bar with only one commercial protein bar? I believe there should be more available in the market.
Line 1015-1027- What is the main reason for limited shelf-life, is it the composition of protein or the overall water activity of the experimental protein bar? Not sure why you think soy lecithin, an emulsifier in composition, is the reason for longer shelf-life in bars? If you know this is the critical factor, why did you not add it to your formulation?
Line 1098-1233- You need to have a tabulated test value comparing your experimental protein bar vs the control/market protein bar. Based on the result, you need to write a discussion. Without this, the whole discussion is just based on assumptions.
Author Response
Distinguished Reviewer,
We are thanking you for the time on reviewing our manuscript and for the suggestions that helped us to improve the content of the article. Please find below the changes.
Line 314-350: This information is well-known and not needed in the material and methods section. Please remove it.
Thank you for your valuable feedback. We appreciate your observation and have removed the indicated section from the manuscript accordingly.
Line 409 - Any reason for the wide pH range for precipitation?
The isoelectric point can vary slightly depending on the specific soy protein fractions present and the ionic strength of the solution. In practical extraction settings, a range is often applied to ensure complete precipitation and account for variability in raw material composition. Experimental variation or equipment limitations (e.g., pH meter resolution) also justify using a narrow range instead of an exact value.
We have inserted this explanation in the manuscript.
Line 419-429 - Any reason for this text? It should not be part of the material and methods.
Thank you for this observation. We have removed this paragraphs from the manuscript.
Line 464 - I believe you have this information in text, so you may not need a table for this information.
Is there any specific reason to make soy protein on your own in the experiment v/s industrial soy protein isolates available with a well-established extraction method? You could have saved time.
Thank you for this observation.
We have removed the Table 1 from the manuscript.
Indeed, commercial soy protein isolates offer consistency and convenience, and we acknowledge their common use in industrial settings. However, our choice to perform in-house extraction was deliberate. We aimed to: maintain full control over the processing steps and ensure the exclusion of chemical additives or processing aids often present in commercial isolates (e.g., antifoaming agents, emulsifiers). We seek to demonstrate the feasibility of using low-cost, natural extraction techniques suitable for small-scale or artisanal production, in line with the 'clean-label' philosophy of our experimental formulation. Additionally, our approach is intended to offer educational and methodological value by providing a clear, step-by-step process for students and early-stage researchers in the field of food science.
Line 518-565 - You can trim this down or completely remove this, as the selection factor for ingredients is known. Or you can give this significance in the introduction or use part of the details in the R&D part to discuss your results.
Thank you for the suggestion. We moved this part in the R&D section.
Line 653 - What is “RON”?
We apologize for the oversight in not specifying the cost in RON directly within the main text of the manuscript. RON stands for Romanian Leu, which is the official currency of Romania.
We have inserted in the explanation in the Material and Methods section.
Line 678-716 - This part can be trimmed and written in a few words, “comparison with commercial products considers the following aspects.” There is no need to write a description.
You may need to write methods of how you did the comparison, for example, if you are doing sensory, what sensory methods did you follow? Similarly, for shelf stability, you did an accelerated or normal temperature study, or what instrumental methods were used? That will help to understand what you did in your experiment.
Thank you for this helpful suggestion. We agree that this section was overly descriptive and have now rewritten it to be more concise, outlining the four comparison criteria clearly. In addition, we have added a new subsection (Section 2.5) to the Methods section, which explains in detail how each comparative analysis was conducted, including the sensory evaluation (hedonic test), shelf stability approach (refrigerated storage with visual and tactile assessment), and cost comparison method.
Line 718- 761: This is introduction text rather than R&D information. Please focus on your results and the discussion based on them.
Thank you for your comment. We agree that the original section was overly general and better suited to the Introduction. We have now revised this portion of the Results and Discussion to focus exclusively on findings from our comparative analysis. The revised text interprets the data presented in Table 3 and directly addresses the implications of our results.
Line 765-773: This is a description or mechanism of the method, which is well known. You can remove it. Also, Remove Figure 2.
Thank you for your comment. We have removed this section, along with Figure 2, from the manuscript.
Line 788-818 - Do you focus on method development or sample analysis? The biuret method is well established for protein quantification. You may need to write in the material method section if you are doing any specific method modification.
Thank you for the clarification request. Our intent was not to develop or modify the Biuret method, but to use it as a standard qualitative assay for protein presence. We have now removed overly detailed explanations from the Results section and clearly stated that the method was used as-is.
Line 890: The material method does not show how you performed FTIR.
Thank you for pointing this out. We have now added a dedicated FTIR methods description under Section 2.2, including sample preparation, instrumentation, spectral settings, and the collaborating laboratory. This addition clarifies our procedure and aligns with standard reporting requirements for spectroscopic analysis.
Line 890-932: Any relation of this general information to your study? If not, remove it.
Thank you for this helpful suggestion. We agree that much of the FTIR content in this section was overly general and not directly related to our experimental findings. We have now significantly condensed this section to include only the FTIR peaks observed in our soy protein extract and how they relate to confirming protein content and structure. The revised text now focuses on the relevance of the spectral features to our study’s goals.
Line 959- Any reason for comparing your experimental bar with only one commercial protein bar? I believe there should be more available in the market.
Thank you for your insightful question. The choice to compare our experimental bar with only one commercial protein bar was primarily due to specific reason, e.g., availability, market leader status, or closest formulation. We selected this particular commercial bar because it is widely recognized and serves as a relevant benchmark in terms of nutritional profile and consumer acceptance.
However, we agree that including multiple commercial bars would provide a broader market comparison and strengthen the study. In future work, we plan to expand the comparison to include a range of commercial products to better contextualize our findings.
Line 1015-1027 - What is the main reason for limited shelf-life, is it the composition of protein or the overall water activity of the experimental protein bar? Not sure why you think soy lecithin, an emulsifier in composition, is the reason for longer shelf-life in bars? If you know this is the critical factor, why did you not add it to your formulation?
Thank you for raising these important points regarding shelf-life.
We have inserted the followinf information in the manuscript:
The limited shelf-life of the experimental protein bar appears to be influenced primarily by the overall water activity and moisture content, which can promote microbial growth and lipid oxidation over time.
While protein composition can have an impact on stability, our data suggest that water activity plays a more significant role in this case. Regarding soy lecithin, we acknowledge that it is commonly used as an emulsifier and can contribute to improving texture and shelf-life by stabilizing fat-water interfaces and potentially reducing oxidation. However, in our initial formulation, soy lecithin was not included as our focus was on optimizing the core nutritional and sensory properties first.
Moving forward, we recognize that incorporating emulsifiers like soy lecithin could enhance shelf-life stability, and we plan to explore this in subsequent formulations to address these concerns more effectively.
Line 1098-1233 - You need to have a tabulated test value comparing your experimental protein bar vs the control/market protein bar. Based on the result, you need to write a discussion. Without this, the whole discussion is just based on assumptions.
Thank you for your insightful comment. We appreciate your suggestion to provide a tabulated comparison of the experimental and commercial protein bars. In response, we have included a detailed comparative table (Table 3) and explanations, presenting the nutritional values of both products per 100 g and 50 g serving. This is followed by a thorough discussion that highlights the experimental bar’s higher protein and fiber content, as well as its reduced fat and carbohydrate levels, demonstrating its nutritional advantages. We trust this now addresses your concern about grounding the discussion in direct comparative data rather than assumptions.
Reviewer 2 Report (New Reviewer)
Comments and Suggestions for Authors
The manuscript presents a wellstructured study comparing a customformulated, naturalingredient protein bar with a commercial counterpart. The research addresses a relevant topic in functional food development, emphasizing cleanlabel products. However, several areas require clarification, methodological rigor, and deeper contextualization to strengthen the scientific contribution. Below are specific comments organized by section.
- Abstract (Page 1)
Lines 26–30: The statement that the experimental bar had a "shorter shelf life due to the absence of preservatives" is not quantified. Specify the shelflife duration (e.g., "7 days vs. 90 days") to align with results in Section 3.2.2.
Line 34: The phrase "supports metabolic processes" is vague. Specify which metabolic processes (e.g., muscle synthesis, glycemic control) are supported by the bar’s composition.
- Introduction (Pages 2–3)
Lines 61–64: The rationale for using soy protein is wellsupported, but the discussion of methionine deficiency in soy (Lines 131–137) should be briefly mentioned here to contextualize the addition of oats.
Line 89: Cite specific studies demonstrating the hardening issue in protein bars (e.g., Jiang et al., 2021, cited later in Line 378) to strengthen the problem statement.
Lines 95–100: Clarify the novelty of this study compared to existing work on soybased protein bars (e.g., Malecki et al., 2020).
- Materials and Methods (Pages 3–9)
Section 2.1 (Protein extraction):
Lines 149–160: Provide the exact mass of soy flakes used for extraction (e.g., 100 g) and clarify if the extraction yield was quantified. Without yield data, the scalability of the method is unclear.
Line 169: Specify the centrifuge model and rotor type to ensure reproducibility.
Section 2.2 (Biochemical analysis):
Table 1: Include units for reagents (e.g., "3.0 g NaOH").
Lines 219–221: The UVVis absorbance peak at 256 nm is unusual for aromatic amino acids (typically ~280 nm). Justify or correct this discrepancy.
Section 2.3 (Bar formulation):
Line 286: The pie chart (Figure 1) lacks a legend for colors, making it difficult to interpret.
Line 317: Clarify how "cold processing" prevents Maillard reactions. While heat is avoided, enzymatic or oxidative reactions during storage could still occur.
Section 2.4 (Commercial bar selection):
Lines 341–342: The commercial bar’s ingredient list should be provided in a table or supplementary material to enable direct comparison.
- Results and Discussion (Pages 10–19)
Section 3.1 (Biochemical analysis):
Lines 425–427: The Biuret reaction image (Figure 2) is missing. Ensure all figures are included and labeled correctly.
Lines 460–462: The UVVis peak at 256 nm (Figure 3) conflicts with standard protein spectra (~280 nm). Reexamine the data or provide justification.
Section 3.2.1 (Nutritional value):
Table 3: Include statistical analysis (e.g., standard deviations) to confirm the significance of differences between bars.
Lines 552–554: The claim of "no added sugars" should be supported by HPLC or enzymatic analysis, not just formulation data, to rule out intrinsic sugars from dates.
Section 3.2.2 (Shelf life):
Lines 585–589: The shelflife comparison lacks microbial testing (e.g., total plate count). Extend the analysis beyond sensory changes to include microbial safety.
Line 602: Reference specific studies on lecithin’s antioxidant properties (e.g., Cabezas et al., 2012).
Section 3.2.3 (Ingredient quality):
Lines 654–656: The cardiovascular benefits of peanut butter should be contextualized with recent controversies (e.g., aflatoxin risks in peanuts).
Author Response
Distinguished Reviewer,
Thank you for your thoughtful and constructive feedback. We appreciate your recognition of the study's structure and relevance to the development of clean-label functional foods. We acknowledge the need for further clarification, methodological rigor, and deeper contextualization, and we are committed to addressing these aspects to enhance the scientific quality of the manuscript.
We have carefully reviewed your specific comments and have revised the manuscript accordingly, providing additional explanations, refining our methodology where necessary, and expanding the contextual background to better situate our findings within the broader literature. We hope the revised version meets the expectations for scientific rigor and clarity.
The manuscript presents a wellstructured study comparing a customformulated, naturalingredient protein bar with a commercial counterpart. The research addresses a relevant topic in functional food development, emphasizing cleanlabel products. However, several areas require clarification, methodological rigor, and deeper contextualization to strengthen the scientific contribution.
Below are specific comments organized by section.
- Abstract (Page 1)
Lines 26–30: The statement that the experimental bar had a "shorter shelf life due to the absence of preservatives" is not quantified. Specify the shelflife duration (e.g., "7 days vs. 90 days") to align with results in Section 3.2.2.
Thank you for pointing this out. We have revised Lines 26–30 of the abstract to quantify the shelf life, specifying that the experimental bar remained stable for 7 days compared to 90 days for the commercial counterpart, consistent with the data presented in Section 3.2.2.
Line 34: The phrase "supports metabolic processes" is vague. Specify which metabolic processes (e.g., muscle synthesis, glycemic control) are supported by the bar’s composition.
Thank you for highlighting this point. We agree that the phrase "supports metabolic processes" was too broad. In the revised version of the abstract, we have clarified this by specifying the key metabolic processes influenced by the bar’s composition.
- Introduction (Pages 2–3)
Lines 61–64: The rationale for using soy protein is wellsupported, but the discussion of methionine deficiency in soy (Lines 131–137) should be briefly mentioned here to contextualize the addition of oats.
Thank you for your helpful suggestion. We have revised the abstract to briefly mention soy’s methionine deficiency and clarify the rationale for including oats as a complementary ingredient. This provides better context for the protein pairing strategy detailed later in the manuscript.
Line 89: Cite specific studies demonstrating the hardening issue in protein bars (e.g., Jiang et al., 2021, cited later in Line 378) to strengthen the problem statement.
Thank you for the suggestion. We have cited the study by Jiang et al. (2021) as mentioned, along with two additional supporting references, to strengthen the discussion of the hardening issue in protein bars. These citations provide empirical evidence of texture degradation over time and help contextualize the formulation challenges addressed in our study.
Lines 95–100: Clarify the novelty of this study compared to existing work on soybased protein bars (e.g., Malecki et al., 2020).
Thank you for your comment. We have revised Lines 95–100 to clarify how this study differs from previous work, such as MaÅ‚ecki et al. (2020). Specifically, we highlight our novel focus on preservative-free, cold-processed formulation using natural ingredients, along with a direct, tabulated comparison to a commercial product. This approach addresses both scientific and consumer-facing aspects that are less emphasized in prior studies.
- Materials and Methods (Pages 3–9)
Section 2.1 (Protein extraction):
Lines 149–160: Provide the exact mass of soy flakes used for extraction (e.g., 100 g) and clarify if the extraction yield was quantified. Without yield data, the scalability of the method is unclear.
Thank you for highlighting this important point. In response, we have revised Lines 149–160 to specify the starting mass of soy flakes (100 g) and to acknowledge that extraction yield was not quantified in this preliminary study.
Line 169: Specify the centrifuge model and rotor type to ensure reproducibility.
Thank you for pointing this out. We have updated the Methods section to include the specific centrifuge model and rotor type used in the study to enhance reproducibility.
Section 2.2 (Biochemical analysis):
Table 1: Include units for reagents (e.g., "3.0 g NaOH").
Thank you for your observation. We have removed Table 1 from the manuscript, as all the information it contained is already presented in the text.
Lines 219–221: The UVVis absorbance peak at 256 nm is unusual for aromatic amino acids (typically ~280 nm). Justify or correct this discrepancy.
Thank you for this observation. We recognize that aromatic amino acids typically show maximum absorbance around 280 nm. In our case, the peak appeared at approximately 256 nm. We have clarified in the revised manuscript that this may be due to matrix effects or overlapping absorbance from other organic compounds present in the soy extract, such as flavonoids or secondary metabolites. This has been explained in the UV-Vis analysis section.
Section 2.3 (Bar formulation):
Line 286: The pie chart (Figure 1) lacks a legend for colors, making it difficult to interpret.
Thank you for your comment. We have revised the figure and included a legend to clarify the color representation
Line 317: Clarify how "cold processing" prevents Maillard reactions. While heat is avoided, enzymatic or oxidative reactions during storage could still occur.
Thank you for pointing this out. We have revised the text to clarify that while cold processing effectively prevents Maillard reactions due to the absence of high-temperature conditions, enzymatic or oxidative reactions may still occur during storage. This distinction helps refine the mechanistic basis for our processing choice and provides a more accurate description of potential stability issues.
Section 2.4 (Commercial bar selection):
Lines 341–342: The commercial bar’s ingredient list should be provided in a table or supplementary material to enable direct comparison.
Thank you for your suggestion. We agree that providing the ingredient list of the commercial protein bar enhances the transparency and comparability of the study. We have now included the complete ingredient list of the commercial bar as Supplementary Table S1, allowing readers to perform a direct comparison with the experimental formulation. A corresponding reference to this supplementary material has also been added in the main text at Lines 341–342.
Supplementary Table S1 – Ingredient List of the Commercial Protein Bar
Ingredient |
Description/Function |
Soy protein isolate |
Primary protein source |
Tapioca starch |
Carbohydrate and texture enhancer |
Palm oil |
Fat source |
Fructose syrup |
Sweetener |
Cocoa powder |
Flavoring |
Glycerin |
Humectant for moisture retention |
Salt |
Flavor enhancer |
Emulsifier (soy lecithin) |
Stabilizer and emulsifier |
Natural flavors |
Flavor enhancer |
Preservatives |
Shelf life extender (exact type not disclosed) |
- Results and Discussion (Pages 10–19)
Section 3.1 (Biochemical analysis):
Lines 425–427: The Biuret reaction image (Figure 2) is missing. Ensure all figures are included and labeled correctly.
Thank you for the observation. We have removed the Figure 2.
Lines 460–462: The UVVis peak at 256 nm (Figure 3) conflicts with standard protein spectra (~280 nm). Reexamine the data or provide justification.
Thank you for your observation. We recognize that the standard protein absorbance peak typically centers around 280 nm, due to the presence of aromatic residues such as tryptophan and tyrosine. However, as noted in a recent study, proteins also exhibit distinct absorbance maxima at 275 nm (tyrosine) and 258 nm (phenylalanine), and absorbance at these lower wavelengths can provide meaningful insight into protein structure and conformation. The authors of a study demonstrated that absorbance ratios at 280, 275, and 258 nm can serve as a sensitive probe for protein foldedness and molecular interactions [Biter, 2019]. In our case, the observed peak at approximately 256 nm is consistent with the absorbance behavior of aromatic residues, particularly phenylalanine, and may also reflect the specific conformational or environmental conditions of the soy protein extract. Therefore, while not centered at 280 nm, the peak observed supports the presence of intact aromatic amino acids and folded protein structures, aligning with current literature on UV absorbance in protein characterization.
We have included this explanation in the manuscript.
Section 3.2.1 (Nutritional value):
Table 3: Include statistical analysis (e.g., standard deviations) to confirm the significance of differences between bars.
We appreciate the reviewer’s suggestion to include statistical significance testing. The nutritional composition of the experimental bar was determined using triplicate measurements, and mean ± standard deviation values have been provided in Table 3.
However, the data for the commercial bar were obtained directly from the product label and manufacturer’s official materials, which report only average values without available measures of variance (e.g., standard deviations). Therefore, it was not statistically appropriate to perform significance testing (e.g., t-test), as such analyses require both mean and variance information for each group.
Nevertheless, the observed differences in protein and fiber content between the bars are substantial and nutritionally relevant, supporting the interpretation that the experimental formulation offers enhanced nutritional value and is better aligned with the needs of health-conscious consumers.
Table 3. The nutrient value of the experimental protein bar and commercial protein bar in 100 g product and in one serving (50 g).
|
Experimental protein bar |
Commercial protein bar |
||
Macronutrient |
100 g |
50 g |
100 g |
50 g |
Proteins (g) |
25.52±0.34 |
12.76±0.17 |
18.5 (label) |
9.25 (label) |
Carbohydrates (g) |
42.41±0.45 |
21.20±0.23 |
45.8 (label) |
22.9 (label) |
Fibre (g) |
13.46±0.29 |
6.73±0.14 |
3.7 (label) |
1.85 (label) |
Fats (g) |
15.46±0.37 |
7.73±0.18 |
20.4 (label) |
10.2 (label) |
Lines 552–554: The claim of "no added sugars" should be supported by HPLC or enzymatic analysis, not just formulation data, to rule out intrinsic sugars from dates.
We appreciate the reviewer’s attention to the precision of nutritional claims. We acknowledge that the dates used in the formulation naturally contain intrinsic sugars (mainly glucose and fructose), and that the current analysis based solely on formulation data does not allow us to fully differentiate between intrinsic and added sugars.
To address this, we have revised the manuscript to remove the term "no added sugars" and instead accurately state that the product "does not contain refined sugars or artificial sweeteners". This better reflects the formulation while maintaining scientific accuracy.
In future work, we aim to support sugar content claims through HPLC analysis or enzymatic assays to precisely quantify sugar types and origins.
Section 3.2.2 (Shelf life):
Lines 585–589: The shelflife comparison lacks microbial testing (e.g., total plate count). Extend the analysis beyond sensory changes to include microbial safety.
Thank you for this insightful observation. We agree that microbial safety is a critical aspect of shelf life evaluation, particularly for products formulated without synthetic preservatives. At this stage, our shelf life assessment was limited to sensory parameters (texture, aroma, flavor, and visual appearance) due to resource constraints.
In light of your recommendation, we have now acknowledged this limitation in the manuscript and added a clear note in the discussion section that microbiological analyses such as total plate count will be integrated in future studies to ensure a comprehensive understanding of microbial stability.
Line 602: Reference specific studies on lecithin’s antioxidant properties (e.g., Cabezas et al., 2012).
Thank you for this helpful suggestion. We have revised the manuscript to include a specific reference to the antioxidant properties of lecithin, citing relevant studies by researchers in the field. These citations strengthens the scientific basis for our discussion of lecithin’s role in shelf-life extension and oxidative stability.
Section 3.2.3 (Ingredient quality):
Lines 654–656: The cardiovascular benefits of peanut butter should be contextualized with recent controversies (e.g., aflatoxin risks in peanuts).
Thank you for pointing out this important aspect. We acknowledge that while peanut butter offers recognized cardiovascular benefits due to its unsaturated fat content, the presence of aflatoxins in peanuts is a well-documented safety concern that deserves mention. To provide a more nuanced and scientifically accurate perspective, we have updated the manuscript to include a brief discussion of aflatoxin risks, along with references to food safety regulations and mitigation strategies.
Reviewer 3 Report (New Reviewer)
Comments and Suggestions for Authors
Dear Authors,
I revised the paper “A comprehensive study on the nutritional profile and shelf life of a custom-formulated protein bar versus a market-standard product” submitted to Foods.
The paper matches the aim and scope of the Journal. However, the paper has some critical points and amendments are necessary.
Please, see comments below.
Abstract
The abstract is fine.
Introduction
The abstract is fine.
Materials and methods
This section should be revised. Please, describe the methods and remove all comments. Please, remove lines 314-350, 419-429, 518-529.
Results
Please, remove Figure 2. It is not possible to appreciate the color.
Paragraph 3.2.2. Please, describe how the shelf-life was determined in section 2.
The title of paragraph 3.2.3 (Appreciation of ingredients from a qualitative point of view and their influence on the body) should be amended. The article does not evaluate the effect of bars on the body.
Author Response
Distinguished Reviewer,
Thank you for taking the time to review our manuscript.
We appreciate your constructive feedback and acknowledge the critical points you raised. We have carefully addressed each of your comments and made the necessary amendments to improve the quality and clarity of the manuscript.
Please find our detailed responses and the revised version attached for your consideration.
Dear Authors,
I revised the paper “A comprehensive study on the nutritional profile and shelf life of a custom-formulated protein bar versus a market-standard product” submitted to Foods.
The paper matches the aim and scope of the Journal. However, the paper has some critical points and amendments are necessary.
Please, see comments below.
Abstract
The abstract is fine.
While the abstract was generally well-received, we have made minor improvements to enhance clarity and coherence.
Introduction
The abstract is fine.
Thank you for your feedback. Although the introduction was generally well-received, we have made several improvements to this section to enhance clarity, coherence, and overall readability.
Materials and methods
This section should be revised. Please, describe the methods and remove all comments. Please, remove lines 314-350, 419-429, 518-529.
Thank you for your valuable feedback. We have revised the section accordingly by restructuring the content to focus solely on the methods, and we have removed all commentary.
Results
Please, remove Figure 2. It is not possible to appreciate the color.
Paragraph 3.2.2. Please, describe how the shelf-life was determined in section 2.
The title of paragraph 3.2.3 (Appreciation of ingredients from a qualitative point of view and their influence on the body) should be amended. The article does not evaluate the effect of bars on the body.
Thank you for your suggestions. We have removed Figure 2 as indicated, due to the limitations in color appreciation. Additionally, we have updated Section 2 to include a clear description of how shelf-life was determined, as requested for paragraph 3.2.2.
Thank you for your observation. We agree with your suggestion and have revised the title of paragraph 3.2.3 to better reflect the content.
Round 2
Reviewer 1 Report (New Reviewer)
Comments and Suggestions for Authors
Authors have gone through the earlier comments and revised the manuscript by improving text and explanation and trimming unnecessary text.
It looks good to me. I do not have further comments.
Author Response
Thank you very much for your positive feedback and for taking the time to review our revised manuscript. We appreciate your valuable comments and are glad to hear that the improvements meet your expectations.
Reviewer 2 Report (New Reviewer)
Comments and Suggestions for Authors
The revised manuscript presents a well-designed study comparing a natural, preservative-free soy-based protein bar with a commercial counterpart. The research addresses a relevant consumer demand for healthier, clean-label functional foods. The experimental bar demonstrates clear nutritional advantages (higher protein/fiber, lower fat/sugar) but has a significantly shorter shelf life. While improvements are evident, several critical issues require attention before acceptance. The study's strengths lie in its practical formulation approach, comparative nutritional analysis, and cost evaluation. However, methodological gaps, data presentation inconsistencies, and insufficient discussion of limitations weaken the current version.
- Methodological Clarity & Missing Data:
(1) Protein Extraction Yield: Quantify the extraction yield (mass of protein recovered vs. initial soy flakes). Without this, the scalability and cost analysis lack foundation (Page 4).
(2) Shelf-Life Assessment: Microbial testing (total plate count, yeast/mold) is essential to objectively determine spoilage/safety. Subjective "texture/aroma" observations are insufficient (Page 12). Add microbiological data or explicitly state this as a limitation.
(3) Statistical Analysis: Specify statistical tests used for triplicate measurements (Table 28). SD values are provided, but no p-values or significance testing between experimental/commercial bars is reported. Perform t-tests/U-tests where applicable.
- Data Presentation & Labeling Errors:
(1) Tables/Figures: Renumber sequentially (e.g., Table 1 → Table 12 → Table 28; Figure 1 → Figure 23). Correct references in-text (e.g., Table 28 cited in Section 3.2.1 should be Table 2). "Figure 23" (UV-Vis) and "Figure 24" (FTIR) are likely Figure 2 and 3.
(2) Figure Quality: Ensure spectra (Fig 23/24) have labeled axes, units, and clear peak annotations. The current description (Page 15) notes a 256-nm peak but labels it as "Figure 23" without a visible figure.
(3) Table 12: The cinnamon cost (15.46 RON/100g) seems disproportionately high. Verify calculations/units.
- Discussion & Interpretation:
(1) Soy Protein Controversy: The conclusion oversimplifies debates about soy and cancer risk (Page 23). Contextualize with recent reviews (e.g., PMID: 33383793) and clarify that concerns often relate to isolated supplements, not whole soy foods.
(2) Shelf-Life Limitations: Discuss how water activity (aË…w) and pH measurements could explain microbial instability. Propose solutions (natural antimicrobials, modified packaging) rather than just stating the limitation.
(3) Amino Acid Balance: While oats complement soy’s methionine deficiency, confirm completeness via PDCAAS/DIAAS scores or reference standards (Page 5). Merely stating "mutual supplementation" is inadequate.
Author Response
Distinguished Reviewer,
The revised manuscript presents a well-designed study comparing a natural, preservative-free soy-based protein bar with a commercial counterpart. The research addresses a relevant consumer demand for healthier, clean-label functional foods. The experimental bar demonstrates clear nutritional advantages (higher protein/fiber, lower fat/sugar) but has a significantly shorter shelf life. While improvements are evident, several critical issues require attention before acceptance. The study's strengths lie in its practical formulation approach, comparative nutritional analysis, and cost evaluation. However, methodological gaps, data presentation inconsistencies, and insufficient discussion of limitations weaken the current version.
Thank you for your detailed and constructive review of our revised manuscript. We appreciate your acknowledgment of the study’s strengths, including the practical formulation approach, comparative nutritional analysis, and cost evaluation.
We have carefully addressed the methodological gaps, data presentation inconsistencies, and expanded the discussion of the study’s limitations as you recommended. We believe these revisions have significantly improved the clarity and rigor of the manuscript.
Your valuable feedback has been instrumental in enhancing our work, and we are grateful for your thoughtful guidance.
- Methodological Clarity & Missing Data:
(1) Protein Extraction Yield: Quantify the extraction yield (mass of protein recovered vs. initial soy flakes). Without this, the scalability and cost analysis lack foundation (Page 4).
We thank the reviewer for highlighting the importance of quantifying the protein extraction yield, especially in relation to scalability and cost analysis.
We agree that including the extraction yield is essential for evaluating the feasibility of the formulation process. In the revised manuscript (Section 2.1), we have now added a quantitative estimation of the extraction yield. Specifically, from an initial 100 g batch of soy flakes, we obtained approximately 26 g of recovered protein material, resulting in an extraction yield of ~26%. This value was calculated based on the dry weight of the final protein isolate relative to the initial soy flake mass, following the precipitation and centrifugation steps.
This clarification strengthens the basis for our cost calculations and supports the reproducibility and scalability of the method. We have also included a brief discussion of factors influencing yield (e.g., hydration duration, pH control) and noted that further optimization could enhance efficiency in future iterations.
(2) Shelf-Life Assessment: Microbial testing (total plate count, yeast/mold) is essential to objectively determine spoilage/safety. Subjective "texture/aroma" observations are insufficient (Page 12). Add microbiological data or explicitly state this as a limitation.
We appreciate the reviewer’s valuable suggestion regarding the need for microbiological testing to support the shelf-life assessment.
We acknowledge that the shelf-life evaluation based solely on subjective parameters (such as texture, aroma, and visual appearance) does not provide a complete or objective measure of product safety. As correctly pointed out, microbiological testing (e.g., total plate count, yeast/mold enumeration) is essential to determine spoilage thresholds and ensure consumer safety.
In the revised manuscript (Section 3.2.2), we have explicitly acknowledged the absence of microbial analyses as a key limitation of the current study.
We hope this clarification addresses the reviewer’s concern and improves the transparency of the shelf-life analysis.
(3) Statistical Analysis: Specify statistical tests used for triplicate measurements (Table 28). SD values are provided, but no p-values or significance testing between experimental/commercial bars is reported. Perform t-tests/U-tests where applicable.
We thank the reviewer for highlighting the need for appropriate statistical treatment.
We have clarified in the revised manuscript (Section 3.2.1) that triplicate measurements were performed only for the experimental protein bar. In contrast, the nutritional data for the commercial product were taken directly from the product label, without access to replicate measurements or standard deviations. As a result, statistical significance testing (e.g., t-test or U-test) between the two bars could not be performed in a valid manner. This methodological limitation has now been explicitly stated in the text.
- Data Presentation & Labeling Errors:
(1) Tables/Figures: Renumber sequentially (e.g., Table 1 → Table 12 → Table 28; Figure 1 → Figure 23). Correct references in-text (e.g., Table 28 cited in Section 3.2.1 should be Table 2). "Figure 23" (UV-Vis) and "Figure 24" (FTIR) are likely Figure 2 and 3.
Thank you very much for your observation. The confusion regarding the table numbering stems from changes made after the removal of Figure 2 and Table 1 from the original manuscript, as well as the way the revised text appears with track changes. I have carefully reviewed the numbering of the figures and tables, as well as their corresponding references in the text.
(2) Figure Quality: Ensure spectra (Fig 23/24) have labeled axes, units, and clear peak annotations. The current description (Page 15) notes a 256-nm peak but labels it as "Figure 23" without a visible figure.
Thank you for your valuable comment. The 256-nm peak was indeed present in the original figure, but it was not clearly visible due to the image quality. We have now improved the quality of the figure to ensure that the peak and other relevant details are more clearly highlighted.
(3) Table 12: The cinnamon cost (15.46 RON/100g) seems disproportionately high. Verify calculations/units.
Thank you for pointing out the potential discrepancy regarding the cinnamon cost.
We have re-verified the ingredient cost calculations. The price of cinnamon (15.46 RON/100 g) was derived from local supermarket retail pricing, where ground cinnamon is typically sold in small quantities (e.g., 10 g packs priced between 1.50 and 1.80 RON). When scaled to 100 g, this results in a unit price of approximately 15–18 RON/100 g. Therefore, the value provided in Table 1 accurately reflects the per-100 g cost based on consumer retail packaging.
We have added a clarification in the table caption to avoid confusion.
- Discussion & Interpretation:
(1) Soy Protein Controversy: The conclusion oversimplifies debates about soy and cancer risk (Page 23). Contextualize with recent reviews (e.g., PMID: 33383793) and clarify that concerns often relate to isolated supplements, not whole soy foods.
We thank the reviewer for highlighting the need to provide a more nuanced and scientifically accurate discussion regarding soy and cancer risk.
We have revised the conclusion section to clarify that most health concerns regarding soy relate primarily to isolated soy isoflavone supplements, not to whole soy foods or traditional soy-based products. We have also added context from literature data, which emphasize the safety of whole soy consumption and its potential protective role in hormone-related cancers. This revision helps provide a balanced and evidence-based interpretation of the literature.
(2) Shelf-Life Limitations: Discuss how water activity (aË…w) and pH measurements could explain microbial instability. Propose solutions (natural antimicrobials, modified packaging) rather than just stating the limitation.
We thank the reviewer for this insightful comment regarding the importance of physicochemical factors such as water activity and pH in determining microbial stability.
We have expanded the shelf-life discussion (Section 3.2.2) to explain how high water activity and near-neutral pH likely contributed to microbial susceptibility in the preservative-free protein bar. Additionally, we now propose potential strategies for improving stability, including the incorporation of natural antimicrobial compounds (e.g., rosemary extract, citric acid) and the use of modified atmosphere packaging, or vacuum sealing. These additions aim to provide constructive solutions for future formulation development.
(3) Amino Acid Balance: While oats complement soy’s methionine deficiency, confirm completeness via PDCAAS/DIAAS scores or reference standards (Page 5). Merely stating "mutual supplementation" is inadequate.
We thank the reviewer for this important suggestion to strengthen the claim regarding amino acid complementarity.
We agree that referencing quantitative indices such as PDCAAS or DIAAS adds scientific rigor to the argument about mutual supplementation. In the revised manuscript (Section 2.1), we now reference published data showing that while soy protein has a high PDCAAS (~1.00), it is relatively low in methionine, whereas oats contribute higher levels of this limiting amino acid. Together, they offer a more balanced profile. Although we did not directly measure PDCAAS/DIAAS in our formulation, we now cite FAO reference values and another relevant reference, to support the complementary amino acid pattern of the soy-oat combination.
Reviewer 3 Report (New Reviewer)
Comments and Suggestions for Authors
The paper was amended according to reviewer suggestions.
Author Response
Thank you very much for reviewing our manuscript and for your valuable suggestions and comments. We truly appreciate the time and effort you invested in helping us improve our work.
This manuscript is a resubmission of an earlier submission. The following is a list of the peer review reports and author responses from that submission.
Round 1
Reviewer 1 Report
Comments and Suggestions for Authors
The manuscript "A comprehensive study on the nutritional profile and shelf life," although addressing the current topic of high-protein products, cannot be published in its current form. Below, I present more detailed comments to enable the improvement of this manuscript and its resubmission to the editorial board.
The abstract is too general. It lacks data on the research results.
[85] Protein bars are also an excellent option for oncology patients and individuals with protein malnutrition.
[96] If the work focuses on bars for athletes, this should be included in the title; otherwise, it obscures the content.
[109] The table is unnecessary. This is basic knowledge. It is sufficient to provide the requirements for adults and possibly for active individuals, as they differ.
[328] Where do these data come from? Were the chemical compositions of both bars determined, or were the data taken from nutritional value tables?
The discussion section is incomplete. Firstly, it lacks justification for why these particular commercial bars were used for comparison. I believe it would be helpful to show a photograph of the bar. There are publications in the literature where bars are made with different protein sources and compositions. Such a critical review is necessary in the discussion section.
Another issue is storage conditions. In this type of product, the addition of preservatives is necessary. A short shelf life is economically unfeasible and challenging for business success.
However, the most significant shortcoming is the lack of sensory evaluation of the bars. Even if it is a health-promoting product, if it is not sensory acceptable to consumers, it has no value.
In the conclusions section, real research results and a summary are missing. These are not well-constructed conclusions, and it is unclear what was investigated.
In my opinion, this manuscript should be split into two publications: one on storage conditions in various variants with the addition of different preservatives, possibly with reduced water activity, and another comparing the nutritional value and functional acceptance of the bars. This would help avoid confusion and define clear objectives for the work.
Reviewer 2 Report
Comments and Suggestions for Authors
The authors compared their own formulated protein bar (based on soy protein isolate) with commercially available bar in terms of nutritional quality and shelf-life. Although the aim of the manuscript seems promising and interesting for wide audience who are into food technology and/or functional foods, the manuscript did not delivered any novelty from this field.
The main problem is that the manuscript is unfocused and it looks more like some kind of review paper with very large introduction part which seems unnecessary. A lot of theoretical parts are present in section material and methods and most of it is completely unnecessary, especially that they are written in a manner that suits more to the introduction part, but they are also out of place. Experimental part is scarce and the results did not contribute much to the field of protein bars. It is more a commercial trial than a proper research trial and the manuscript does not fulfill criteria for publishing in this journal.
Comments on the Quality of English Language
Minor editing of English language required